# Vinpocetine Ameliorates Metabolic-Syndrome-Associated Bladder Overactivity in Fructose-Fed Rats by Restoring Succinate-Modulated cAMP Levels and Exerting Anti-Inflammatory Effects in the Bladder Detrusor Muscle

**DOI:** 10.3390/biomedicines10112716

**Published:** 2022-10-26

**Authors:** Wei-Chia Lee, Hong-Ren Yu, You-Lin Tain, Kay L.H. Wu, Yao-Chi Chuang, Julie Y.H. Chan

**Affiliations:** 1Division of Urology, Kaohsiung Chang Gung Memorial Hospital, Chang Gung University College of Medicine, Kaohsiung 833, Taiwan; 2Department of Paediatrics, Kaohsiung Chang Gung Memorial Hospital, Chang Gung University College of Medicine, Kaohsiung 833, Taiwan; 3Institute for Translational Research in Biomedicine, Kaohsiung Chang Gung Memorial Hospital, Kaohsiung 833, Taiwan

**Keywords:** fructose, metabolic syndrome, overactive bladder, succinate, vinpocetine

## Abstract

Succinate and its receptor, the G protein-coupled receptor 91 (GPR91), have pathological implications in metabolic syndrome (MetS) and its associated bladder dysfunction, particularly in decreasing bladder cAMP levels and promoting proinflammation. Using fructose-fed rats (FFRs), a rat model of MetS, we investigate the effects of vinpocetine (a phosphodiesterase-1 inhibitor) and celecoxib (a selective cyclooxygenase-2 inhibitor) on MetS-associated bladder overactivity. Phenotypes of the overactive bladder, including increased micturition frequency and a shortened intercontractile interval in cystometry, were observed in FFRs, together with elevated succinate levels in the liver and serum and the downregulation of GPR91 in the liver and urinary bladder. Treatments with vinpocetine and celecoxib improved tissue fibrosis and ameliorated the overexpression of the inflammatory cytokines, such as IL-1β, in the liver and bladder. In bladder organ bath studies, vinpocetine, but not celecoxib, treatment restored the contraction and relaxation responses of the detrusor muscle strip in response to KCl, carbachol, and forskolin stimulation. At a molecular level, vinpocetine and celecoxib treatments modulated the downstream messengers of GPR91 (i.e., ERK1/2 and JNK), suppressed NF-κB and IL-1β expressions in the bladder, and prevented the fibrogenesis observed in FFRs. The exogenous application of succinate to a bladder organ bath significantly reduced the forskolin-induced cAMP production by the detrusor muscle, which was notably restored in the presence of vinpocetine. Together, these results suggest that vinpocetine may alleviate the MetS-associated bladder overactivity by restoring the succinate-modulated detrusor cAMP production and exerting the anti-inflammatory effects in the bladder detrusor muscle.

## 1. Introduction

Metabolic syndrome (MetS) is characterized by central obesity and insulin resistance combined with hypertension, hypertriglyceridemia, and hypercholesterolemia [1]. With the increasing consumption of a Western diet (e.g., a high fructose diet) and sedentary lifestyle, the prevalence of MetS has rapidly increased worldwide [1,2]. Overactive bladder (OAB) syndrome is a medical condition which is defined by a cluster of complex symptoms, including urgency, urinary frequency, and nocturia, with or without urgent incontinence [3]. The high global prevalence of OAB causes a public health burden due to the detrimental effects on the quality of life and productivity of patients, particularly when the effective treatments for specific subtypes of OAB remain elusive [3,4]. Accumulative evidence suggests that MetS and OAB could share common pathophysiologies, such as dysregulated nutrient-sensing pathways (e.g., insulin resistance in the bladder mucosa [5] and excessive succinate intake [6]), chronic ischemia, and proinflammatory status [7]. Furthermore, the MetS-associated OAB is an important subtype of OAB, which could not be well managed by nonspecific pharmacotherapies (e.g., antimuscarinics, β-3 agonist, and botulinum toxin injections) [4].

Succinate, known as an intermediary metabolite of tricarboxylic acid (TCA) cycle, is involved in several pathological conditions through activating the G protein-coupled receptor 91 (GPR91) [8]. In animal models and human studies of MetS, increased circulating succinate is associated with hypertension, ischemia heart disease, and obesity-related metabolic disturbance [9,10,11]. One of the prominent pathological implications in endogenous succinate overproduction is hepatic cirrhosis and proinflammatory status [12]. Furthermore, male lower urinary tract symptoms may be developed due to the dysregulated glucose metabolism in MetS and excessive metabolites from TCA cycle [13]. In vitro cell culture studies [14,15] further suggested that succinate, via its action on the GPR91, could inhibit the forskolin-induced cellular cyclic adenosine monophosphate (cAMP) production in the urothelial cells, and could counteract with the mirabegron (a β3 adrenergic receptor agonist used in OAB treatment)-induced cAMP production in bladder detrusor muscle cells. In addition, excessive succinate intake could promote bladder dysfunction (e.g., bladder fibrosis and a decrease of bladder capacity) and enhance metabolic perturbations in rats [6].

Vinpocetine is a phosphodiesterase (PDE) 1 inhibitor which is approved for clinical use in stroke or dementia [16]. This PDE1 inhibitor relaxes bladder detrusor muscles by increasing the tissue cAMP levels [17], which underpins its therapeutic use for OAB treatment. Vinpocetine is the only available agent for clinical purposes [17]. In this regard, close to 60% of the OAB patients showed an improvement in clinical symptoms and/or urodynamic parameters after treatment with vinpocetine for 2 weeks [18]. More recently, vinpocetine has been reported to be a potent anti-inflammatory agent which prohibits nuclear factor (NF)-κB transcription by blocking the IκB kinase complex [19].

Based on the epidemiological observation, the increasing prevalence of MetS is associated with the consumption of fructose [2]. At the same time, a high-fructose diet induces the traits of MetS. High-fructose-fed rats (FFRs) are used as a common animal model of MetS [20] that present the phenotype of MetS-associated bladder overactivity [5,21]. During the process of glucose metabolism [22], a high-fructose diet would induce excessive succinate production [12,23]. We therefore explore in the present study the effects of vinpocetine on the MetS-associated bladder overactivity in FFRs and the role of succinate-modulated cAMP levels and tissue inflammation in the effect of vinpocetine on the bladder detrusor muscle.

## 2. Materials and Methods

### 2.1. Animals

This study was conducted in accordance with the guidelines of the National Research Council, USA. The experimental protocol was approved by the Institutional Animal Ethics Committee (permit number: 2019032203). All surgery was performed under anesthesia, and every effort was made to minimise the suffering of the animals and the number of animals used in our experiments.

Eighty female Wistar rats (BioLASCO Taiwan Co., Ltd., Taipei, Taiwan; weight: 200–240 g) were randomly allocated to 4 groups (*n* = 20) and subjected to an experimental course of 12 weeks. They were maintained in a facility accredited by the Association for Assessment and Accreditation of Laboratory Animal Care International under controlled temperatures (24 °C ± 0.5 °C) and a light–dark cycle of 12 h each. For our experiments, the rats were divided into 4 groups, namely the control (regular chow), fructose (fructose group; fructose-rich diet; 60% fructose diet, Harlan Teklad, Madison, WI, USA), fructose plus vinpocetine (V6383/Lot#118H46803, Sigma-Aldrich, Darmstadt, Germany) (vinpocetine group; 1 mg/kg) for tube feeding delivery through the mouth from the ninth week [18], and fructose plus celecoxib (Pfizer, New York, NY, USA) (celecoxib group; 20 mg/kg for tube feeding delivery through the mouth from the ninth week), as a treatment control group. Dosages of vinpocetine and celecoxib used were adopted from studies that used the drugs for the same purpose [24,25].

### 2.2. Metabolic Cage Study and Oral Glucose Tolerance Test (OGTT)

Twelve rats in each group were placed in individual 3701M081 metabolic cages (Tecniplast, Buguggiate, Italy) at the end of week 11, as reported previously [5]. After a familiarization period (24 h), the volume of liquid consumed, micturition frequency, and urine output were measured for 3 days, and an average value was determined. Subsequently, the OGTT was performed after an overnight fast [5].

### 2.3. Measurements of Blood Pressure, Filling Cystometry, and Metabolic Parameters

Twelve rats in each group were weighed and then anaesthetized using subcutaneous urethane (1.2 g/kg). Polyethele-50 catheters were placed in the left carotid artery to measure arterial pressure by a PowerLab^®^ 16S system with a P23 1D transducer (Gould-Statham, Oxnard, CA, USA) [26]. The mean arterial pressure (MAP) was calculated using the formula MAP = 1/3 systolic pressure + 2/3 diastolic pressure. Through the urethra, the bladder catheter was connected using a T-tube to a pressure transducer and a microinjection pump (CH-4103; Infors, Bottmingen, Switzerland). Room-temperature saline was infused into the bladder at a rate of 0.08 mL/min. The voiding pressure was recorded using an RS3400 chart recorder (Gould, Cleveland, OH, USA). All of the rats were observed for a minimum period of 30 min to ensure that the voiding pattern was stable. Subsequently, reproducible micturition cycles were recorded for 1-h periods and used for evaluation [5,26]. An overdose of urethane was then injected to sacrifice the rats. Blood samples were collected for metabolic parameters by dry chemistry. The succinate level was assessed by the colorimetric Assay kit (k649-100, BioVision, Waltham, MA, USA).

### 2.4. Liver and Urinary Bladder Histological Study

To characterize the morphological changes in the livers and urinary bladders of the rats, the left lobe of liver and bladder base of each group were fixed in formalin, embedded in paraffin, and stained with the standard Masson’s trichrome stain. Images of the staining in each section were recorded using a digital camera (EasyScan, Motic, Hong Kong, China), and their intensities were determined by the use of image analysis software (Image-J; National Institutes of Health, Bethesda, MD, USA).

### 2.5. Concentrations-Related Contractions (CRCs)

As in our previous report [27,28], two strips of mucosa-intact detrusor muscle (10 × 2 mm and weighing 10 to 15 mg) from the dorsal part of the bladder body in each bladder were used (*n* = 8, for each group). The bladder strips were attached to a force-displacement transducer and mounted in a Compact Organ bath system (Panlab, SLU, Barcelona, Spain) containing 10 mL of physiological saline solution. The resting tension in the tissues was controlled at 1 g. CRCs to KCl (Sigma-Aldrich) (10 mM to 300 mM); carbachol (Sigma-Aldrich) (0.1 μM to 10 μM) or ATP (Sigma-Aldrich) (10 μM to 3 mM) were obtained in a stepwise manner after the response to the previous concentration had reached a plateau. The contraction responses were recorded as tension (g) per cross-sectional area (mm^2^) for analysis.

### 2.6. In Vitro Relaxation Responses to Forskolin and Vinpocetine

Meanwhile, we carried out the tests of relaxation responses to forskolin (F3917/Lot#SCB20653, Sigma-Aldrich) and vinpocetine. After the resting tension on the tissues was adjusted at 1 g, the mucosa-intact bladder strips were soaked in carbachol (3 μM) to stimulate a moderate contraction of the strips. Then, cumulative concentration–response curves to forskolin (0.1 μM to 300 μM) were recorded in these precontracted bladder strips. After washing out and re-equilibrium, we performed the relaxation response to vinpocetine (0.1 μM to 30 μM) to carbachol (3 μM) precontracted strips.

### 2.7. Determination of cAMP Levels in Bladders

For determining the amount of cAMP levels in naïve mucosa-denuded detrusor muscles, we used the Direct cAMP ELISA kit (ADI-900-066, Enzo Life Science Inc., Farmingdale, NY, USA), according to the manufacture’s protocol, for the frozen tissues of the bladders in each group (*n* = 8). In another study, we also measured the cAMP amount of the mucosa-intact bladder tissue under forskolin with or without succinate (S3674/Lot#SLBX6330, Sigma-Aldrich) stimulation for each group (*n* = 8). We divided tissues from a bladder into four portions and put into warmed and oxygenated physiological saline solutions in an organ bath. Two of them were stimulated for 20 min with succinate (30 μM or 100 μM) and the following forskolin (30 μM) stimulation for 20 min. The other two strips were also prepared in the organ bath in the absence or in the presence of forskolin (30 μM) stimulation. After this interval, the tissues were immediately frozen in liquid nitrogen. The cAMP amounts of these bladder tissues were measured by ELISA. The assays were performed in duplicate, and the pellet weights were used to normalize the data.

### 2.8. Western Blots for Liver and Bladder Proteins

Western immunoblotting was performed using the liver tissues and denuded detrusor muscles for GPR91, proinflammatory mediators, and fibrosis markers. The procedures were previously reported [5,28]. In brief, alternative samples from each group were homogenized on ice in CelLytic^TM^MT cell lysis buffer (Sigma-Aldrich) containing a protease inhibitor. The total protein was measured using the Pierce 660-nm protein assay (ThermoFisher, Waltham, MA, USA). Sodium dodecyl sulphate-polyacrylamide gel electrophoresis was performed using the Laemmli buffer system.

Antibodies raised against GPR91 (1:5000; NBP-92180, Novus, Denver, CO, USA), extracellular signal-regulated kinases (ERK) 1/2 (1:5000; 4695S, Cell Signal, Danvers, MA, USA), c-Jun N-terminal kinase (JNK) (1:5000; 9252S, Cell Signal), Cyclooxygenase (COX)-2 (1:2000; ab15191, Abcam, Cambridge, UK), NF-κB (1:2000; 8242S, Cell Signal), tumour necrosis factor (TNF)-α (1:2000, 3707S, Cell Signal), interleukin (IL)-1β (1:500; ab9787, Abcam), IL-6 (1:500; ab9324, Abcam), transforming growth factor (TGF)-β1 (1:5000; WH147355, Abclonal, Woburn, MA, USA), α-smooth muscle actin (SMA) (1:10,000; 67735-1, Proteintech, Chicago, IL, USA), Collagen I (1:500; ARG21965, Arigo, Taipei, Taiwan), Collagen III (1:500; ARG20786, Arigo), Fibronectin (1:2500; 610077, BD, Franklin Lakes, NJ, USA), and GAPDH (1:10,000; MAB374, Millipore, Billerica, MA, USA) were used. The proteins of GPR91 may express in different molecular weights among different organs [29].

### 2.9. Statistical Analysis

All data are presented as the mean ± standard error of means (SEM). Before all of the statistical analyses were performed, the Shapiro–Wilk test was used to confirm that the data were compliant with normal distribution. Data were subjected to one-way analysis of variance (ANOVA) and multiple comparisons by using the Bonferroni test. For all statistical tests, *p* < 0.05 was considered statistically significant.

## 3. Results

### 3.1. General Characteristics and Micturition Behaviour

Table 1 and Figure 1 present the general characteristics, micturition behaviour, and biochemical results of all experimental groups (*n* = 12 in each group). No adverse events were found throughout the experiments. Compared with normal diet controls, the fructose-fed group showed higher mean values in MetS traits, including higher MAPs, hypertriglyceridemia, hypercholesterolemia, impaired OGTT, and homeostasis model assessments of insulin resistance (HOMA-IR). The FFRs also exhibited the phenotype of bladder overactivity, including increased micturition frequency in the metabolic cage study and a shortened intercontractile interval in cystometry. The fructose/vinpocetine group exhibited significantly more MetS traits (i.e., higher MAPs, hypercholesterolemia, elevated HOMA-IR, and impaired OGTT) than the control group, but no significant change in bladder overactivity. The fructose/celecoxib group also exhibited some MetS traits (i.e., higher MAPs, hypercholesterolemia, and impaired OGTT), but no bladder overactivity. Figure 1C shows a significant decrease of cAMP levels in the mucosa-denuded detrusor muscle of the fructose-fed group when compared with the controls. The reduction in cAMP production in the detrusor muscles of the FFRs was significantly reserved by the vinpocetine treatment, but not the celecoxib treatment. These results indicate that a high fructose diet may promote the MetS-associated OAB.

### 3.2. The Association among a Higher Hepatic Succinate Level, the Proinflammatory Status, and Liver Fibrosis in the FFRs

The liver is the key organ wherein fructose is transformed into succinate through the TCA cycle. We therefore investigated changes in hepatic succinate levels as well as in the hepatic functions of the FFRs. Figure 2 shows a significant increase in hepatic succinate levels, along with decreased expression of their receptor, GPR91, in the livers of all of all FFRs when compared with the controls. Furthermore, significant increases of inflammatory mediators (i.e., TNF-α, IL-6, and IL-1β) were observed in the fructose-fed group and the fructose/vinpocetine group (i.e., TNF-α and IL-6), but not in the fructose/celecoxib group (Figure 2C–E). In the Masson trichrome stain of the liver (Figure 3A,B), the zone-three perisinusoidal fibrosis (hepatic fibrosis stage one in non-alcohol steatohepatitis (NASH)) [30] and lipid steatosis were observed in all of the FFRs. Moreover, the fructose-only group showed fibrous portal expansion (hepatic fibrosis stage two) in liver sections (Figure 3B). At the same time, a significant overexpression of fibrogenesis markers, including TGF-β1, SMA, and collagen I, was noted in the fructose-only group, but not in either of the treatment groups. Together, these results suggest that a high fructose diet increases hepatic succinate levels and evokes hepatic inflammation and liver fibrosis that could be ameliorated by the addition of vinpocetine or celecoxib treatments.

### 3.3. The Association between an Increase of Circulating Succinate Level, the Proinflammatory Status of the Bladder, and Bladder Fibrosis in the FFRs

Our observation of an increase of succinate levels in the liver implies that this metabolite could be an active intermediate responsible for the damage of bladder tissues and the OABs in the FFRs. As shown in Figure 4, a significant increase of succinate levels in serum, but not in the bladder tissue, was detected in all of the FFRs when compared with the controls. The downregulation of GPR91 was observed in the bladder of all of the fructose-fed groups. At the same time, overexpressions of ERK1/2, JNK, NF-κB, IL-1β, IL-6, TNF-α, and COX-2 were noted in the fructose-only group, but not in either treatment group. In addition, the vinpocetine- and celecoxib-treated groups showed significantly decreased expressions of NF-κB and IL-1β. Figure 5A illustrates the submucosal bladder fibrosis in the fructose-only group by Masson trichrome stain, which was appreciably reduced by both treatments. For fibrogenesis markers, the fructose-only group showed significant increases in TGF-β1, collagen I, collagen III, and fibronectin expression (Figure 5B–E). Moreover, except for collogen I, the overexpressions of all other fibrogenesis markers were significantly restored in the fructose/vinpocetine group. No significant change in fibrogenesis markers was observed in the fructose/celecoxib group.

### 3.4. The Effects of Vinpocetine and Celecoxib on the Reduced Bladder Detrusor Contractility in the FFRs

To further identify the influence of fructose on the contractility of the bladder detrusor muscle and the protection conferred by vinpocetine, in vitro experiments on bladder strips were performed. Figure 6A–C illustrates the concentration-dependent contractile responses of the mucosa-intact bladder’s smooth muscle strips of all of the groups in response to the application of KCl, carbachol, or ATP. In this study, contractile responses to KCl, carbachol, and ATP were normalized to the 300 mM KCl-induced contraction. Compared with the control group, the contractile responses to KCl and carbachol, but not ATP, were significantly blunted in the fructose-only group. Both treatments exerted appreciable protection on the reduced contractility of the detrusor muscle in the FFRs, with vinpocetine appearing to be more prominent than celecoxib.

Tissue cAMP levels play an active role in the relaxation of the detrusor muscle; as such, foskolin, a cell-permeable diterpene that directly activates adenylyl cyclase for the production of cAMP, was used to elucidate the engagement of cAMP on the protective effect of vinpocetine. As shown in Figure 6D, forskolin stimulation evoked a dose-dependent relaxation of the mucosa-intact bladder strips that was precontracted in the presence of carbachol (3 μM). Compared with the normal-diet control group, the fructose-only group showed a significant impairment in forskolin-stimulated detrusor relaxation, which was partially protected by vinpocetine but not by celecoxib. It is of interest to note that the addition of vinpocetine improved the action of celecoxib which caught up with the improvement exerted in the fructose/vinpocetine group (Figure 6E).

### 3.5. Tissue cAMP Production in Response to Forskolin and Modulation by Succinate in the Bladder Detrusor Muscle

Finally, the cAMP levels in the bladder tissues of all of the groups were quantified. As shown in Figure 6F, forskolin (30 μM) stimulation produced an increase of cAMP production in the mucosa-intact bladder strips of all of the groups, of which the increase was significantly lesser in the fructose-fed group but notably increased in the fructose/vinpocetine group. In addition, after the organ bath was preincubated with succinate (30 or 100 μM), the forskolin-stimulated cAMP production in all of the groups was significantly inhibited. Of note, the amount of forskolin-induced cAMP in the presence of succinate remained higher in the fructose/vinpocetine group. In contrast, the celecoxib treatment had no protective effect on the reduced cAMP production in the fructose-fed group.

## 4. Discussion

The results of the present study are interpreted to suggest that vinpocetine may ameliorate MetS-associated bladder overactivity by restoring the succinate-modulated detrusor cAMP production and suppressing the tissue inflammation and fibrogenesis of the bladder. The hypothesized mechanisms of the molecular events between the succinate overproduction induced by fructose intake and the MetS-associated bladder dysfunction are illustrated in Figure 7. Our study has several strengths. First, we demonstrated that high-fructose diet increases both hepatic and circulating succinate levels that could facilitate the proinflammatory status, hepatic fibrosis, and bladder fibrosis. Second, our results support the hypothesis that excessive circulating succinate may deteriorate the bladder function, inhibit the bladder’s cAMP production, induce proinflammation, impair detrusor function, and enhance bladder fibrosis, all of which may lead to the MetS-associated OABs in rats. Third, we report that vinpocetine may alleviate MetS-associated OABs via its dual actions of restoring tissue cAMP levels and promoting anti-inflammation. The underlying protective action of vinpocetine on the metabolic inflammation of the bladder was further supported by celecoxib-treated results. Taken together, the current study highlights the negative impact of excessive circulating succinate on the MetS-associated bladder overactivity in FFRs. Under vinpocetine treatment, the FFRs improved their bladder detrusor properties in the contractility and relaxation function as well as their micturition behaviour, which were mediated via the restoration of the detrusor cAMP levels, the suppression of the proinflammation status, and the subsequent amelioration of the fibrogenesis of the bladder.

Excessive fructose intake induces overproduction of succinate in the liver of NASH [12,23]. Succinate, via activation on its receptor GPR91, functions as a signalling molecule to cause hepatic fibrosis by activating hepatic stellate cells [31,32,33]. In this process, both the increase in fibrogenesis markers (e.g., α-SMA and collagen I) and in pro-inflammatory cytokines (e.g., IL-6 and TNF-α) have been demonstrated [31,32]. The results of the current study are in good agreement with the previous studies, and demonstrate significant increases of succinate levels (20.4 to 28.5 μM) in the liver of the high-fructose diet groups. The half-maximal effective response for GPR91 is obtained with succinate concentrations at 28–56 μM [34]. The downregulation of liver GPR91 in FFRs could be a compensatory reaction to the chronic high succinate stimulation. Subsequently, the increase of proinflammatory mediators (i.e., TNF-α, IL-6, and IL-1β) and fibrogenesis markers (i.e., TGF-β1, α-SMA, and collagen I), as well as hepatic fibrosis in NASH stage two, were observed in the fructose group. All of these molecular and tissue responses to the high-fructose diet were appreciably alleviated by vinpocetine and celecoxib treatments that reduced the hepatic fibrosis of the FFRs to NASH stage one.

The elevated serum succinate levels and the downregulation of bladder GPR91 in all of the fructose groups suggest that circulating succinate may function as a key mediator that acts on the GPR91 to promote the downstream effectors, including the inhibition of cAMP production and the promotion of tissue inflammation in the bladder. Elevated circulating succinate levels have been reported to be associated with cardiovascular diseases in rodents and humans [9,11]. In this study, the decrease of detrusor cAMP levels in the fructose group could be a consequence of the dysregulated succinate-GPR91 signalling. Mossa et al. [15] reported that succinate via GPR91 inhibits cAMP production in bladder smooth muscle cells, suggesting an inhibitory role of succinate on cAMP-mediated bladder relaxation. In the current study, we found that the increase in cAMP production stimulated by forskolin in the mucosa-intact detrusors of the control rats was significantly blunted by the addition of succinate, providing evidence to support the argument of an inhibitory action of succinate on cAMP production. Our observations further suggest that the protection conferred by vinpocetine treatment on the bladders of the FFRs may be attributed to the preservation of tissue cAMP levels, in addition to its anti-inflammatory action. The vinpocetine treatment not only augmented the forskolin-induced cAMP production in the bladders of the FFRs, but also preserved appropriate cAMP levels in the detrusor muscle strips of the FFRs in the presence of exogenous succinate. At the same time, vinpocetine in the organ bath augmented the relaxation of the bladder strips from the fructose group. Our observations that the protective effect of celecoxib, a specific COX 2 inhibitor, on functional impairments in the relaxation of the detrusor muscle was notably enhanced with the addition of vinpocetine lend support to the importance of restoring detrusor cAMP in the treatment of MetS-associated OABs.

Vinpocetine could also improve the succinate-associated bladder overactivity in FFRs by exerting an anti-inflammatory effect. Vinpocetine has been demonstrated as a potent anti-inflammatory agent by directly inhibiting IKK activity and suppressing NF-κB expression [19]. Metabolic perturbations promote chronic systemic low-grade inflammation [35]. Chronic low-grade inflammation, along with obesity and insulin resistance, have been speculated to play a central role in the pathogenic mechanism of MetS [24,25,35]. Recently, the role of succinate as an inflammatory signal had been emphasized [36]. Succinate-GPR91 signalling activates downstream effectors, including ERK 1/2, JNK, NF-κB, COX2, and IL-1β [36], to promote inflammation. In cultured urothelial cells, Mossa et al. [14] reported that succinate via GPR91 activates the second messengers of the mitogen-activated protein kinase pathway (i.e., ERK and JNK) and elicits the overexpression of inducible nitric oxide synthase (iNOS). In our previous study [26], the increase of iNOS expression in the bladder mucosa of FFRs was associated with bladder overactivity. In the current study, both vinpocetine and celecoxib treatments modulated inflammatory mediators to inhibit the pathological molecular patterns, particularly in the suppression of NF-κB and IL-1β overexpression in the detrusor muscles of the FFRs.

Modulating and suppressing inflammatory mediators in the detrusor muscle may benefit MetS-associated OABs by preventing detrusor fibrosis and bladder dysfunction. Flores et al. [6] reported that excessive succinate intake caused bladder fibrosis and impaired detrusor contractility in rats. Our results are consistent with these findings, as they demonstrate the elevated succinate levels, bladder fibrosis, and impaired detrusor contractility in the fructose group. Metabolic inflammation may result in bladder fibrosis, which progresses to chronic inflammation, repeated wound healing, sustained collagen deposition, and extracellular matrix remodelling, gradually increasing the stuffiness of the bladder [7]. In this study, we found that both vinpocetine and celecoxib treatments provided anti-inflammatory effects to restore the fibrogenesis and detrusor function in FFRs. Rahnamai et al. [37] reported the roles of prostaglandins and PDE in the development of proinflammatory OABs and suggested treatment with COX-2 inhibitors and PDE inhibitors for OABs. Accordingly, celecoxib was included in the present study as a reference treatment for OABs [38,39]. The inclusion of celecoxib in the experimental design also helped to consolidate the conclusion on the dual protective actions of vinpocetine on MetS-associated OABs via the anti-inflammation and preservation of cAMP in the detrusor muscles (Figure 7).

Our study has several limitations. First, metabolic perturbations could deteriorate bladder function through several mechanisms, such as pelvis ischemia, neuropathy, insulin resistance, and excessive succinate production [7]. In the present study, we focussed on the protective actions of vinpocetine on the succinate-modulated bladder dysfunction of the deficiency in detrusor cAMP and proinflammation. Vinpocetine is a pleiotropic agent that possesses a variety of pharmacological actions, including increasing tissue cGMP levels, vasodilation, anti-oxidation, and the protection of injury-induced vascular remodelling [19]. As such, a complete understanding of the mechanisms underlying bladder protection by vinpocetine remains to be elucidated. Second, the complexity of succinate-related pathophysiology is still unclear [8]. Therefore, different origins of succinate production (i.e., endogenous succinate from the mitochondria of the liver or exogenous succinate from the metabolites of microbiota) might have different influences on animals [10,12,40]. Further preclinical and clinical studies are required to validate the effectiveness of vinpocetine in such a wide range of pathological conditions in MetS. Third, in compliance with the Three-R principle for the use of laboratory animals, the treatment of vinpocetine to control the animals on the normal diet was not included in the present study. In this regard, treatment with a low dose PDE4 inhibitor has been demonstrated to exert minimal effects on the bladder function of normal rats [41].

## 5. Conclusions

Our study results demonstrate the potential of vinpocetine in treating MetS-associated OABs by restoring the succinate-modulated detrusor cAMP levels and exerting anti-inflammatory effects in the detrusor muscle, contributing to the beneficial effect of the PDE inhibitor on the micturition behaviours and detrusor functions in FFRs. Our results also unveil the underlying pathogenesis of succinate overproduction on MetS-associated inflammation and fibrosis in liver and urinary bladders. These data would lay the foundation of further investigation for the therapeutic potential of PDE1 inhibitors in the treatment of MetS-associated bladder dysfunctions.

## Figures and Tables

**Figure 1 biomedicines-10-02716-f001:**
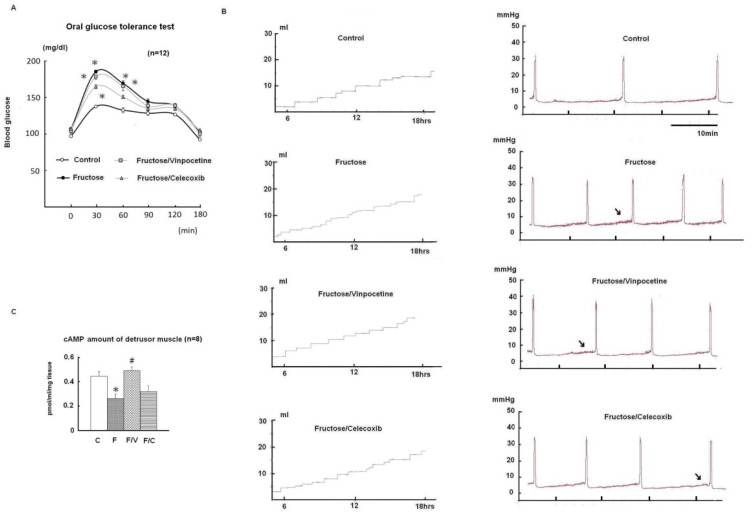
Glucose tolerance, micturition behaviour, and mucosa-denuded detrusor cellular cyclic adenosine monophosphate (cAMP) amounts of experimental animals. (**A**) The response curves of plasma glucose to oral glucose tolerance test, (**B**) metabolic cage study (left) and filling cystometry (right) among study groups, (**C**) the cAMP amounts of mucosa-denuded detrusor muscle in different groups. Data are represented in mean ± SEM. One-way ANOVA with Bonferroni tests were carried out (* shows significance in comparison with the control group; ^#^ shows significance in comparison with the fructose-only group). Arrows indicate the increased basal tone of cystometry. C: control; F: fructose; F/V: fructose/vinpocetine; F/C: fructose/celecoxib.

**Figure 2 biomedicines-10-02716-f002:**
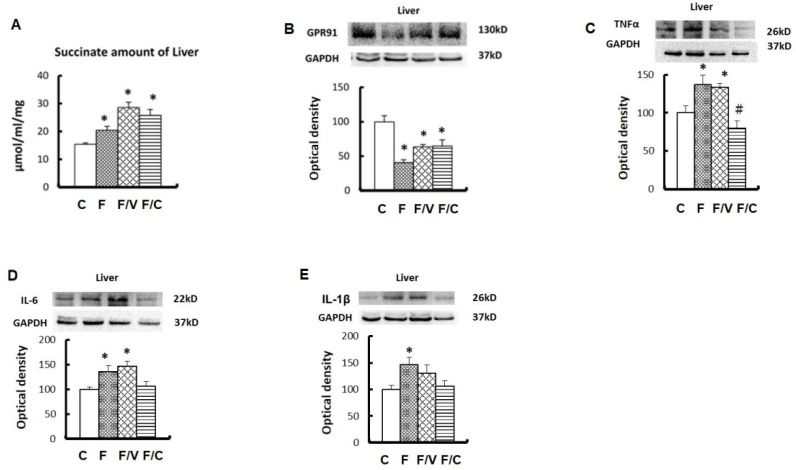
The contents of succinate in the liver tissues, the expressions of the GPR91 protein, and the expressions of proinflammatory mediators, such as TNF-α, IL-6, and IL-1β, among the groups. The succinate levels in the liver tissues were evaluated by ELISA (**A**)**.** Representative Western blot gels and densitometric analyses of proteins in GPR91 (**B**), TNF-α (**C**), IL-6 (**D**), and IL-1β (**E**) among the study groups. Western blots data are expressed in percentage to the control group denoted as 100%. Data are presented in mean ± SEM, *n* = 8 in each group. * shows significance in comparison with the control group, and # with the fructose group, by one-way ANOVA with Bonferroni test.

**Figure 3 biomedicines-10-02716-f003:**
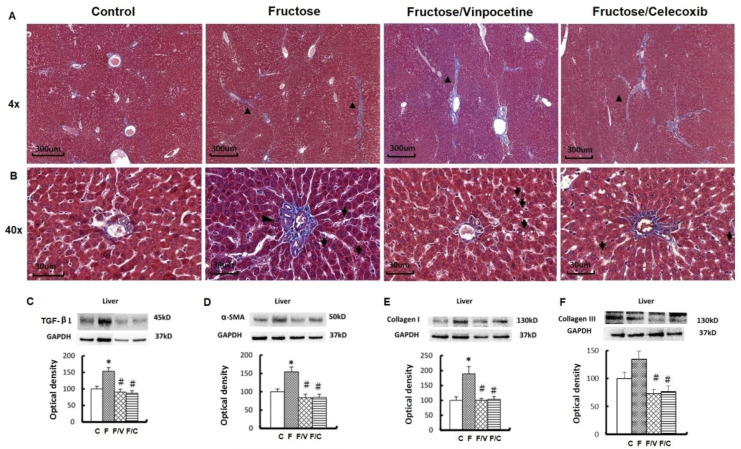
The effects of long-term fructose feeding on hepatic fibrosis, lipid steatosis, and fibrogenesis markers. (**A**) Masson’s trichrome stain of liver tissues in different groups. Arrowhead: the zone-3 perisinusoidal fibrosis. Reduced from 4×. (**B**) Masson’s trichrome stain of liver tissues in different groups. Arrowhead indicates fibrous portal expansion in the fructose-only group. Arrow: lipid steatosis in all fructose groups. Reduced from 40×. (**C**–**F**) Western blots of TGF-β1, α-SMA, collagen I, and collagen III, respectively, of liver tissues among groups. Data are expressed as mean ± SEM. *n* = 8. * *p* < 0.05 versus controls and # with versus fructose group.

**Figure 4 biomedicines-10-02716-f004:**
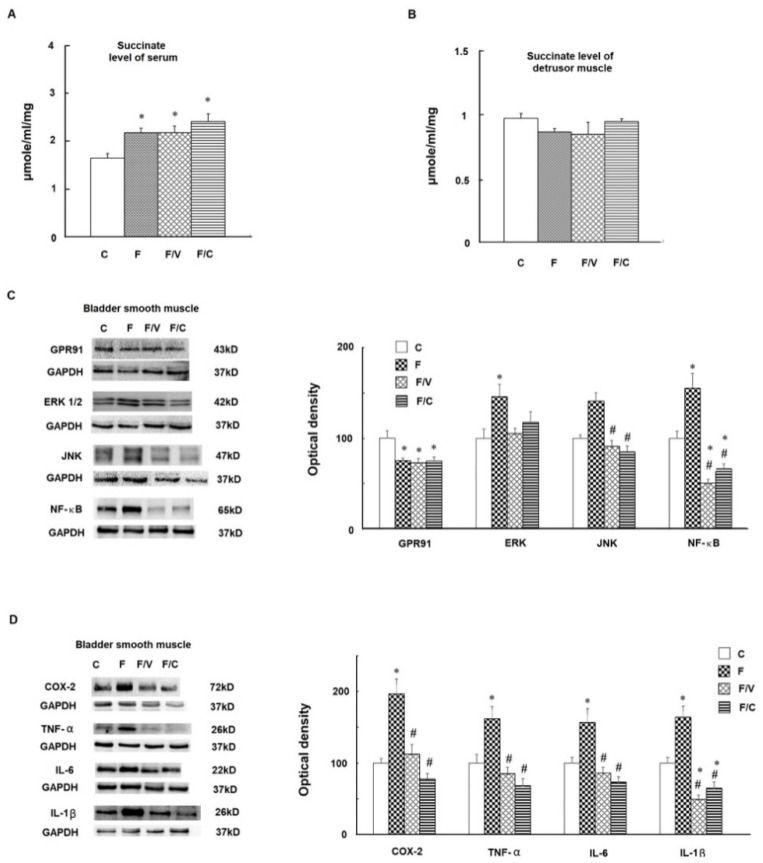
The content of succinate in serums and bladders and the protein expressions of GPR91 and its downstream effectors in the bladders of different groups. (**A**). Succinate levels of serums. (**B**) Succinate levels of detrusor muscles. The succinate levels were evaluated by ELISA. Western blots of GPR91, ERK 1/2, JNK, NF-κB (**C**), COX-2, TNF-α, IL-6, and IL-1β (**D**). Data are expressed as mean ± SEM. *n* = 8. * *p* < 0.05 versus controls and # versus fructose group.

**Figure 5 biomedicines-10-02716-f005:**
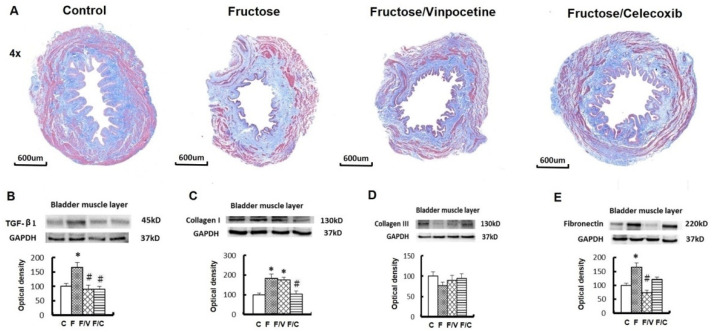
Urinary bladder fibrosis and fibrogenesis markers in experiment groups. (**A**) Masson’s trichrome stain of bladder sections in different groups. The fructose-only group presents obvious submucosal fibrosis. Western blots of fibrogenesis markers in the bladder, including TGF-β1 (**B**), collagen I (**C**), collagen III (**D**), and fibronectin (**E**). Data are expressed as mean ± SEM. *n* = 8. * *p* < 0.05 versus controls and # versus fructose group.

**Figure 6 biomedicines-10-02716-f006:**
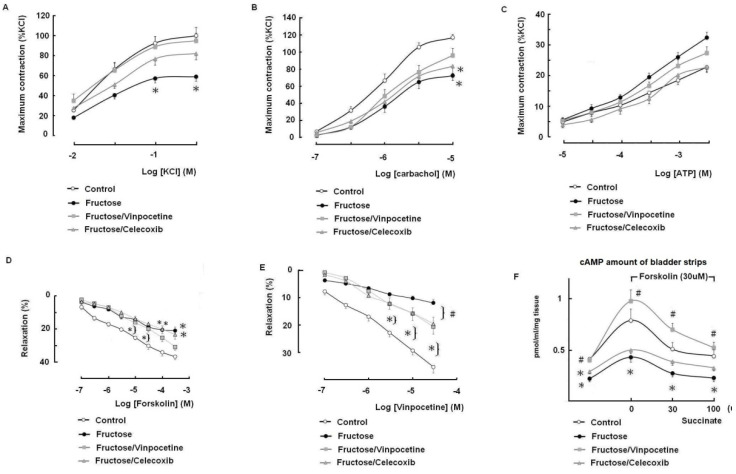
The response curves of the concentrations-related detrusor contractions by using KCl (**A**), carbachol (**B**), and ATP (**C**); the curves of relaxation of the detrusor muscles in response to forskolin (**D**) and vinpocetine (**E**); and the detrusor cAMP levels under forskolin and succinate stimulation (**F**) among the groups. Mucosa-intact bladder strips were used in these studies. Each point represents the mean ± SEM of observations (*n* = 8). One-way ANOVA with Bonferroni test: * significant difference compared with the controls (*p* < 0.05) and # versus fructose group.

**Figure 7 biomedicines-10-02716-f007:**
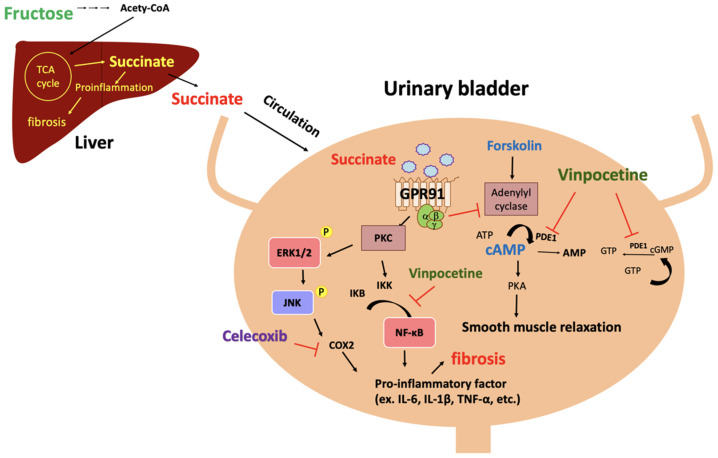
The hypothetic mechanisms of excessive fructose intake raising circulating succinate levels and the following molecular events through GPR91 activation in the bladder. Two negative impacts of succinate/GPR91 activation are concerned: inhibiting the production of detrusor cAMP and eliciting proinflammation along with bladder fibrosis.

**Table 1 biomedicines-10-02716-t001:** General characteristics and micturition behaviour of experimental animals.

	Control	Fructose	Fructose/ Vinpocetine	Fructose/ Celecoxib
**General characteristics**				
Body weight (gm)	282.9 ± 4.1	289.7 ± 4.6	287.0 ± 6.1	293.2 ±3.7
Bladder weight (mg)	105.3 ±2.0	107.1 ± 2.3	108.3 ± 3.3	108.8 ± 2.2
MAP (mmHg)	127.6 ± 2.4	151.5 ± 3.2 *	135.5 ± 1.5 *	153.8 ± 3.1 *
**Fasting biochemistry parameters**				
Triglycerides (mg/dL)	41.8 ± 3.18	78.0 ± 4.49 *	47.3 ± 3.32	42.8 ± 3.26
Cholesterol (mg/dL)	53.3 ± 1.38	85.3 ± 2.97 *	85.9 ± 2.17 *	91.6 ± 2.27 *
Hemoglobin A_1c_	4.05 ± 0.03	4.08 ± 0.03	4.11 ± 0.02	4.04 ± 0.02
Glucose (mM)	5.4 ± 0.11	5.8 ± 0.17	5.9 ± 0.19	6.0 ± 0.17
Insulin (mU/L)	12.4 ± 1.12	24.4 ± 1.6 4 *	20.1 ± 0.77 *	12.2 ± 0.8
HOMA-IR	3.0 ± 0.23	6.4 ± 0.42 *	5.2 ± 0.15 *	3.3 ± 0.24
**Metabolic cage study/24 h**				
Water intake (mL)	35.9 ± 1.2	33.1 ± 1.2	36.3 ± 1.4	35.4 ± 1.9
Urine output (mL)	23.1 ± 0.56	22.3 ± 1.2	22.8 ± 1.1	24.8 ± 1.6
No. voids	17.1 ± 1.0	21.9 ± 0.7 *	17.5 ± 0.6	18.1 ± 0.9
**Cystometric parameters**				
Voiding pressure (mmHg)	24.6 ± 0.7	26.6 ± 1.6	25.8 ± 0.7	25.6 ± 1.3
ICI (min)	13.6 ± 0.7	8.8 ± 0.3 *	11.4 ± 0.7	11.9 ± 0.6

Data are presented as mean ± S.E.M. One-way ANOVA and Bonferroni post-hoc test. * indicates significance in comparison with the control group. MAP: mean arterial pressure, HOMA-IR: homeostasis model assessment of insulin resistance, ICI: inter-contractile interval.

## Data Availability

The data presented in this study are available on reasonable request from the corresponding author.

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
