# Peer review of "Vinpocetine Ameliorates Metabolic-Syndrome-Associated Bladder Overactivity in Fructose-Fed Rats by Restoring Succinate-Modulated cAMP Levels and Exerting Anti-Inflammatory Effects in the Bladder Detrusor Muscle"

_biomedicines, 2022, doi:10.3390/biomedicines10112716_

Round 1

Reviewer 1 Report (Previous Reviewer 1)

The revised form of paper titled (Vinpocetine Ameliorates Metabolic syndrome-associated Bladder Overactivity in Fructose-fed Rats by Restoring Succinate-Modulated cAMP Level and Exerting Anti-inflammatory Effects in the Bladder Detrusor Muscle) was much improved compared to the original form. I would like to ask the authors to do some more revisions:

1- Intro: the aim of the work should be better explored and the following senetence especially needs rewriting "with particular focus 91 on the protection against the succinate-modulated cAMP level and tissue inflammation in 92 the bladder detrusor muscle."

2- I recommend if the authors can find a way to presnet figure 4 in a more clear form

3- Explain this part and describe how you quantified the stains???

"The color setting and the image-associated quantification were determined by 141 using image analysis software"

Kindly send the revised version NOT in track change, send it with highlights on the new additions only.

Author Response

Reviewer 1

Comments and Suggestions for Authors

The revised form of paper titled (Vinpocetine Ameliorates Metabolic syndrome-associated Bladder Overactivity in Fructose-fed Rats by Restoring Succinate-Modulated cAMP Level and Exerting Anti-inflammatory Effects in the Bladder Detrusor Muscle) was much improved compared to the original form. I would like to ask the authors to do some more revisions:

We thank the Reviewer for the affirmative views on the revised manuscript, and appreciate the opportunity to further improve on our manuscript. Per instruction by the Editorial Office, all changes are highlighted in yellow background in the revised manuscript.

  1. Intro: the aim of the work should be better explored and the following sentence especially needs rewriting "with particular focus on the protection against the succinate-modulated cAMP level and tissue inflammation in the bladder detrusor muscle."

Response: We thank the Reviewer for the comment. The sentence is now revised to “We therefore explored in the present study the effects of vinpocetine on the MetS-associated bladder overactivity in FFRs, and the role of succinate-modulated cAMP level and tissue inflammation in the effect of vinpocetine on bladder detrusor muscle.” (P. 2, Lines 79-82).

  1. I recommend if the authors can find a way to present figure 4 in a more clear form.

Response: Thanks for this suggestion. We have revised the figure 4 to increase its readability. The revised figure now appears on P. 8.

  1. Explain this part and describe how you quantified the stains??? "The color setting and the image-associated quantification were determined by using image analysis software".

Response: We thank the Reviewer for this comment. We revised this part and reported that “Image of the staining in each section was recorded using digital camera (EasyScan, Motic, Hong Kong, China), and its intensity was determined by the use of image analysis software (Image-J; National Institutes of Health, Bethesda, MD, USA).” The narrations now appear on P. 3, Lines 126-131.

Reviewer 2 Report (Previous Reviewer 3)

The authors have addressed the comments adequately.

Author Response

We thank the Reviewer for the affirmative views on our work.

Reviewer 3 Report (Previous Reviewer 2)

The article was successfully revised, and I recommended acceptance in the present form.

Author Response

We thank the Reviewer for the affirmative views on our work.

Round 2

Reviewer 1 Report (Previous Reviewer 1)

thanks

This manuscript is a resubmission of an earlier submission. The following is a list of the peer review reports and author responses from that submission.

Round 1

Reviewer 1 Report

Pape titled (Vinpocetine Ameliorates Succinate-associated Bladder Overactivity in Fructose-fed Rats by Restoring Detrusor cAMP Amount and Exerting Anti-inflammatory Effects) by Lee et al. demonstrated that vinpocetine successfuly protected from bladder overactivity in rats suffering from IR due yo fructose feeding. Authors claimed the mechanism is mediated by restoring cAMP and reduction of the inflammatory burden.

Although authors performed good efforts in performing this study, this paper contains many deffects that prevents its acceptance for publication in Biomedicines, first of all:

1- title does not express the content!! succinate was used in fructose fed rats? or in vitro study

2- Experiment control groups are lacking as vinpocetine control group 

3- what is the value of the fructose+celecoxib group? no value actually.

4- Statistical analysis is not correct: Authors have to check the normality of distribution of the results by a suitable post hoc test (such as Shapiro-Wilk test or K-S test) before deciding to choose certain ANOVA. If the normality test indicated normal dist of the data, so use one-way ANOVA, if not, use non parametric ANOVA.

5- Source of animals must be declared.

6- Authors should give the source of chemicals, kits and antibodies completely and consistently (code, company, town, state and country) & version for software

7- Data should be presented as mean+-SD (not SE) this is as authors do not cover the universe for this study.

Author Response

Paper titled (Vinpocetine Ameliorates Succinate-associated Bladder Overactivity in Fructose-fed Rats by Restoring Detrusor cAMP Amount and Exerting Anti-inflammatory Effects) by Lee et al. demonstrated that vinpocetine successfully protected from bladder overactivity in rats suffering from IR due to fructose feeding. Authors claimed the mechanism is mediated by restoring cAMP and reduction of the inflammatory burden.

Although authors performed good efforts in performing this study, this paper contains many defects that prevents its acceptance for publication in Biomedicines, first of all:

  1. title does not express the content!! succinate was used in fructose fed rats? or in vitro study

Response: Thanks for the reviewer’s comment. Our responses are as follows.

  1. With regard to the reviewer’s concerns, we may revise our title as the following, “Vinpocetine ameliorates metabolic syndrome-endogenous succinate-associated bladder overactivity in fructose-fed rats by restoring detrusor cAMP amount and exerting anti-inflammatory effect.” We do report the high concentrations of succinate in the liver (Figure 2A) and in serum (Figure 4A) of the fructose-fed rats.
  2. In the preliminary study, we had performed the fructose/succinate group. However, we found this group cannot generate a linear dose-related response to the pathology of the bladder as well as the results of organ bath studies by adding on succinate intake. That kinds of results had been demonstrated by Flores et al [Ref 1], in which the Dahl rats plus succinate did not induce an overwhelmed pathology compared to the Dahl rats. It may attribute to the property of succinate/GPR91 signalling in its half-maximal effective response between 28 and 56μM in succinate concentrations [Ref 2]. And, the maximum concertation of succinate in physiological response of succinate/GPR91 signalling should be at 180μM [Ref 3, 4]. In addition, the complexity of succinate-related pathophysiology is still unclear [Ref 5]. The different origins of succinate production (i.e., endogenous succinate from mitochondria of liver or exogenous succinate from metabolites of microbiota) might have different influence on animals. [Ref 6-8] We have addressed this issue in Page 31, line 5 to line 10
  3. We do demonstrate the counteraction between forskolin and succinate (30μM and 100μM) on the detrusor cAMP production in vitro, as illustrated in Figure 6F.

References

  1. Flores, M.V.; Mossa, A.H.; Cammisotto, P.; Campeau, L. Succinate decreases bladder function in a rat model associated with metabolic syndrome. Urodyn. 2018, 37, 1549–1558.
  2. He, W. Miao, F.J-P. Citric acid cycle intermediates as ligands for orphan G-protein-coupled receptors. Nature 2004, 429, 189-193.
  3. McCreath, K.J., Espada, S. Targeted disruption of the SUCNR1 metabolic receptor leads to dichotomous effects on obesity. Diabetes 2015, 64, 1154-1167.
  4. Mossa, A.H.; Flores, M.V.; Nguyen, H.; Cammisotto, P.G.; Campeau, L. Beta-3 Adrenoceptor Signaling Pathways in Urothelial and Smooth Muscle Cells in the Presence of Succinate. Pharmacol. Exp. Ther. 2018, 367, 252-259.
  5. Grimolizzi, F., Arranz, L. Multiple faces of succinate beyond metabolism in blood. Haematologica 2018, 103, 1586-1592.
  6. Liu, X.-J.; Xie, L. Succinate-GPR-91 receptor signalling is responsible for nonalcoholic steatohepatitis-associated fibrosis: Effects of DHA supplementation. Liver Int. 2020, 40, 830-43.
  7. Serena, C., Ceperuelo-Mallafré, V. Elevating circulating levels in human obesity are linked to specific gut microbiota. ISEM J 2018, 12, 1642-1657.
  8. Fernández-Veledo, S., Vendrell, J. Gut microbiota-derived succinate: Friend or foe in human metabolic diseases? Rev Endocr Metab Disord 2019, 20, 439-447.

2- Experiment control groups are lacking as vinpocetine control group 

Response: Thanks for the reviewer’s comment. We have considered this issue in the stage of study design. However, the vinpocetine was launched more than 30 years. [Ref 1] The safety profile of vinpocetine has been extensively discussed. We added sentences as follows, “In a short-term human study, no obvious adverse effect of vinpocetine was reported [Ref 2]. In addition, FDA warmed that vinpocetine should not be used during pregnancy, which may hurt the baby or result in miscarriage [Ref 3].” Page 8, Line 1-4. According to the principle of reduction of animal use, we discarded this arm in the current study.

References

  1. Zhang, Y.-S.; Li, J.-D.; Chen. Y. An update on vinpocetine: new discoveries and clinical implications. Eur J Pharmacol 2018, 819, 30-4.
  2. Meador, K.J., Leeman-Markowski. Vinpocetine, cognition and epilepsy. Epilepsy Behav 2021, 119, 107988.
  3. Commissioner, Office of the (2019-06-03). "Statement on warning for women of childbearing age about possible safety risks of dietary supplements containing vinpocetine". FDA. Retrieved 2019-06-04.

3- what is the value of the fructose+celecoxib group? no value actually.

Response: Thanks for the reviewer’s comment. As we known, metabolic syndrome can elicit several metabolic perturbations and affect organ dysfunction. And, the vinpocetine is a pleiotropic agent. In the preliminary study, we found the vinpocetine treatment can improve the pathology of non-alcoholic steatohepatitis as well as liver function by its anti-inflammatory effects. Therefore, there would be the possibility that vinpocetine improve the bladder dysfunction of rats by its anti-inflammatory effect on the liver or bladder, not relate to the PDE1 inhibition. As shown in Figure 5A, the anti-inflammatory effect of celecoxib improved the bladder fibrosis of fructose-fed rats. Whether the restored detrusor cAMP amount of vinpocetine-treatment group resulted from the PDE1 inhibition or the restored detrusor muscle volume? Using the fructose/celecoxib group as a treatment control, we can figure out the roles of vinpocetine in treating bladder dysfunction by anti-inflammation and PDE1 inhibition. In addition, another reviewer considered that results of the celecoxib treatment group should be comprehensively discussed.

4- Statistical analysis is not correct: Authors have to check the normality of distribution of the results by a suitable post hoc test (such as Shapiro-Wilk test or K-S test) before deciding to choose certain ANOVA. If the normality test indicated normal dist. of the data, so use one-way ANOVA, if not, use non parametric ANOVA.

Response: We agree the reviewer’s viewpoint. Usually, the n=20 in human studies or n=6-8 in animal studies may roughly fit the requirement of normality assumption in statistics. It is because the experimental rat is similar to each other. Most of researchers stood on the shoulders of giants to design their studies. In our study, 3 kinds of data sets are available, including physiological data, data of concentrations-related contraction, and optical density of Western blots. We believe that physiological data of animals and data from organ bath studies can fit the assumption of normality. With regard to the semi-quantitative entity of Western blot, we can find values obtained from blots merged in a narrow range in the same group, which lead to not violating the normality. By the reviewer’s suggestion, we test our data sets by the Shapiro-Wilk test and no data set violate the assumption of normality. The examples were list below. A. triglyceride for biochemistry data test. B. KCl-induced detrusor contractility in 300mM. C. optical density in GPR91. We add a sentence as follows, “The normality of data was assessed by Q-Q plot and Shapiro-Wilk test by using Wizard pro.” Page 17, Line 4-5.

5- Source of animals must be declared.

Response: Thank for this comment. The rats used in the current study were bought from BioLASCO Taiwan Co., Ltd., Taipei, Taiwan. This issue had been disclosed in Page 9, Line 11.

6- Authors should give the source of chemicals, kits and antibodies completely and consistently (code, company, town, state and country) & version for software

Response: Thank for this suggestion. We have updated the information in the section of materials and methods. (Page 10, Line 3 to Line 5; Page 11, Line 14 to Line 15; Page 13, Line 3 to Line 4 and Line 11 to Line 12; Page 14, Line 7-8 and Line 11; Page 16, line 1 to Line 14).

7- Data should be presented as mean+-SD (not SE) this is as authors do not cover the universe for this study.

Response: Thank for this comment. However, we have the different opinion. In epidemiological studies, the data in a study group may have a wide range of distribution, such as the value of age, blood pressure, and so on. Therefore, researchers should report the standard deviation to disclose the distribution of these data. However, in the animal studies, the experimental rat is highly similar and conditioned to each other, which results in a narrow range of data distribution in nature, as shown in figures for question 4. Under such circumstances, researchers would concern the significant difference of items among experimental groups rather than the data distribution. Using standard error of means, researchers and readers are more convenient to compare the difference of means among groups in experiments than using the standard deviation. Hence, we would like to keep to present our data by using standard error of means.

Reviewer 2 Report

Congratulations for the present study! 
The theme is very actual and responds to a lot of urodynamic questions.

I recommended minor revision for some English errors and phrase constructions.

Also it would be helpful if you can include a strength and limitations paragraph.

Author Response

Comments and Suggestions for Authors

Congratulations for the present study! 
The theme is very actual and responds to a lot of urodynamic questions.

We thank the Reviewer’s comment.

  1. I recommended minor revision for some English errors and phrase constructions.

Response: Thanks for the reviewer’s suggestion. We have re-checked our manuscript.

  1. Also, it would be helpful if you can include a strength and limitations paragraph

Response: Thanks for this suggestion. We added a paragraph for the limitations of this study at the end of discussion section. 

Our study has some limitations. In MetS, metabolic perturbations can deteriorate bladder function through several mechanisms, such as pelvis ischemia, neuropathy, insulin resistance, and excessive succinate production [Ref 1]. In this study, we used vinpocetine to target on the pathology of succinate-associated bladder dysfunction, including the deficiency of detrusor cAMP and bladder fibrosis induced by proinflammation. In fact, vinpocetine is a pleiotropic agent and has a variety of pharmacological targets, including increasing cAMP and cGMP levels in tissue, vasodilation, anti-oxidation, anti-inflammation, and antagonized injury-induced vascular remodeling [Ref 2]. Vinpocetine might elicit synergistic therapeutic effects on those multifactorial MetS-associated bladder dysfunction as presenting in the current study. Indeed, we are unable to explore the whole spectrum of vasculopathy or neuropathy in MetS and the probable effects of vinpocetine in a single study [Ref 3, 4]. In addition, the complexity of succinate-related pathophysiology is still unclear [Ref 5]. The different origins of succinate production (i.e., endogenous succinate from mitochondria of liver or exogenous succinate from metabolites of microbiota) might have different influence on animals. [Ref 6-8] Further preclinical and clinical studies are needed to validate the effectiveness of vinpocetine in such a wide range of pathological conditions in MetS. (Page 30, Line 8 to Page 31, Line 10)

References

  1. Hsu, L.-N.; Hu, J.-C. Metabolic syndrome and overactive bladder syndrome may share common pathophysiologies. Biomedicines 2022, 10, 1957.
  2. Zhang, Y.-S.; Li, J.-D.; Chen. Y. An update on vinpocetine: new discoveries and clinical implications. Eur J Pharmacol 2018, 819, 30-4.
  3. Zhang, L., Yang, Li. Anti-inflammatory effects of vinpocetine in atherosclerosis and ischemia stroke: A review of the literature. Molecules 2015, 20, 335-347.
  4. Wada, N., Karnup, S. Current knowledge and novel frontiers in lower urinary tract dysfunction after spinal cord injury: Basic research perspectives. Urol Sci 2022, 33, 101-113.
  5. Grimolizzi, F., Arranz, L. Multiple faces of succinate beyond metabolism in blood. Haematologica 2018, 103, 1586-1592.
  6. Liu, X.-J.; Xie, L. Succinate-GPR-91 receptor signalling is responsible for nonalcoholic steatohepatitis-associated fibrosis: Effects of DHA supplementation. Liver Int. 2020, 40, 830-43.
  7. Serena, C., Ceperuelo-Mallafré, V. Elevating circulating levels in human obesity are linked to specific gut microbiota. ISEM J 2018, 12, 1642-1657.
  8. Fernández-Veledo, S., Vendrell, J. Gut microbiota-derived succinate: Friend or foe in human metabolic diseases? Rev Endocr Metab Disord 2019, 20, 439-447.

Reviewer 3 Report

In the manuscript entitled “Vinpocetine ameliorates succinate-associated bladder overactivity in fructose-fed rats by restoring detrusor cAMP amount and exerting anti-inflammatory effects” the authors investigated the effects of vinpocetine or celecoxibMetS-associated bladder overactivity. Below are the suggestions to improve the manuscript.

1.    How did the authors arrive at the optimum dosages of vinpocetine and celecoxib?

2.    Did the authors try vinpocetine+celecoxib combination in their experiment? What would be the expected results? The authors should discuss.

Author Response

In the manuscript entitled “Vinpocetine ameliorates succinate-associated bladder overactivity in fructose-fed rats by restoring detrusor cAMP amount and exerting anti-inflammatory effects” the authors investigated the effects of vinpocetine or celecoxib on MetS-associated bladder overactivity. Below are the suggestions to improve the manuscript.

  1. How did the authors arrive at the optimum dosages of vinpocetine and celecoxib?

Response: For animal studies, we usually evaluated the effects of medicine at the maximum dosage for humans. Sometimes, we would tapper the dosage used in previous animal studies. As we known, the metabolic syndrome is a prediabetes status. Hence, a relatively low dosage may work for rats with metabolic syndrome. We provided the references, listing below and addressing references in Page 10, Line 5 and line 7.

  1. Meador, K.J., Leeman-Markowski. Vinpocetine, cognition and epilepsy. Epilepsy Behav 2021, 119, 107988.
  2. Chan, P-C., Liao, M-T., Hsieh, P-S. The dualistic effect of COX2-mediated signaling in obesity and insulin resistance. Int J Mol Sci 2019, 20, 3115.
  3. Liu, T.-T.; Shih, K.-C. Importance of cyclooxygenase 2-mediated low-grade inflammation in the development of fructose-induced insulin resistance in rats. Chin J Physiol 2009, 52, 65-71.
  4. Did the authors try vinpocetine+celecoxib combination in their experiment? What would be the expected results? The authors should discuss.

Response: Thanks for this comment. The role of celecoxib group was used as a treatment control to define the anti-inflammatory effect on the bladder dysfunction of FFRs. We do not perform vinpocetine plus celecoxib treatment in FFRs. However, we observed the improvement of traits of metabolic syndrome in the celecoxib group. We added a comment as below, “Chronic low-grade inflammation along with obesity and insulin resistance have been speculated to play a central role in the pathogenic mechanism of MetS. [Ref 1, 2] We cannot ignore the fact that the COX-2 inhibitor may have its role in treating the inflammation associated with MetS. In perspective, the combination of vinpocetine and celecoxib might provide their therapeutic potential against the non-alcoholic steatohepatitis and OAB induced by MetS.” (Page 29, line 16 to Page 30, Line 6)

References

  1. Chan, P-C., Liao, M-T., Hsieh, P-S. The dualistic effect of COX2-mediated signaling in obesity and insulin resistance. Int J Mol Sci 2019, 20, 3115.
  2. Liu, T.-T.; Shih, K.-C. Importance of cyclooxygenase 2-mediated low-grade inflammation in the development of fructose-induced insulin resistance in rats. Chin J Physiol 2009, 52, 65-71.

Round 2

Reviewer 1 Report

Unfourtnately, authors did not reply to most of the previous comments specially those concerning study design (did not convince me on the importance of their selection) or statistical analysis. This paper does not add value to the scientific community & does not deserve actually to be published in Biomedicines.